# Therapeutic Effect of Natural Products and Dietary Supplements on Aflatoxin-Induced Nephropathy

**DOI:** 10.3390/ijms25052849

**Published:** 2024-03-01

**Authors:** Ebenezer Ofori-Attah, Mai Hashimoto, Mayu Oki, Daisuke Kadowaki

**Affiliations:** 1Faculty of Pharmaceutical Sciences, Sojo University, 4-22-1 Ikeda, Nishi-Ku, Kumamoto 860-0082, Japan; mhashi@ph.sojo-u.ac.jp (M.H.); zzuouzz836@gmail.com (M.O.); 2DDS Research Institute, Sojo University, 4-22-1 Ikeda, Nishi-Ku, Kumamoto 860-0082, Japan

**Keywords:** aflatoxin-B1, kidney, oxidative stress, antioxidant, natural product, dietary supplement

## Abstract

Aflatoxins are harmful natural contaminants found in foods and are known to be hepatotoxic. However, recent studies have linked chronic consumption of aflatoxins to nephrotoxicity in both animals and humans. Here, we conducted a systematic review of active compounds, crude extracts, herbal formulations, and probiotics against aflatoxin-induced renal dysfunction, highlighting their mechanisms of action in both in vitro and in vivo studies. The natural products and dietary supplements discussed in this study alleviated aflatoxin-induced renal oxidative stress, inflammation, tissue damage, and markers of renal function, mostly in animal models. Therefore, the information provided in this review may improve the management of kidney disease associated with aflatoxin exposure and potentially aid in animal feed supplementation. However, future research is warranted to translate the outcomes of this study into clinical use in kidney patients.

## 1. Introduction

Aflatoxins are a group of highly toxic metabolites produced by the fungi *Aspergillus flavus* and *Aspergillus parasiticus* [1,2]. They occur naturally, and are estimated to contaminate 25% of cereals worldwide [3]. Under optimum temperature conditions (24–27 °C) and humidity >62%, these fungi can grow and produce aflatoxin on all agricultural commodities [4,5]. The degree of contamination varies depending on geographical location, food storage procedures, and processing methods. It is abundant in developing countries where safety regulations for agricultural commodities are often ignored [6,7].

Aflatoxins are well known to be potent mutagens, hepatotoxic, hepatocarcinogenic, nephrotoxic, teratogenic, genotoxic, and immunosuppressive agents that inhibit various metabolic processes, causing liver, kidney, and heart damage [6,8,9,10]. In all animal species, the most affected organs with aflatoxins are the liver and kidneys [11,12,13]. Aflatoxins are metabolized in the liver, producing highly reactive chemical intermediates that cause free radical production, lipid peroxidation, and cell damage [14,15,16,17,18]. Aflatoxins and its metabolites are exposed to different parts of the nephron, causing nephrotoxicity before it is excreted in the urine. Aflatoxin-induced toxicities have been associated with kidney weight loss, glomerular basal membrane thickening, decreased glomerular filtration rate, decreased urine flow, and apoptosis in animal studies [19,20,21,22,23].

Oxidative stress is associated with the progression of kidney disease [24]. Research has demonstrated that oxidative stress is a critical risk factor for aflatoxin toxicity, and that exposure to aflatoxin raises contents of reactive oxygen species (ROS), which can impair cellular redox homeostasis, leading to oxidative stress-induced kidney injury [25,26]. Most complications of chronic kidney disease (CKD), including inflammation and cardiovascular disease (CVD), the leading cause of mortality in CKD patients, have been associated with elevated levels of oxidative stress [27,28,29,30]. As a result, reducing oxidative stress is emphasized as a useful way to treat aflatoxin nephrotoxicity.

Despite significant advancements in kidney disease treatment, effective medications are still few. The high cost of medication, along with the severe side effects, has led to the search for novel natural remedies, particularly those produced from plants and spices [31]. Research on natural remedies for renal disease prevention and treatment has gained popularity due to their antioxidant and anti-inflammatory properties [32], which can alleviate toxin-induced nephrotoxicity and prevent tissue damage [20,21,22,23]. To counteract the adverse effects of aflatoxin-induced nephrotoxicity, various dietary antioxidant supplements, natural products, and remedies have been tested for their antifungal, anti-aflatoxigenic, and antioxidant activities. Many studies have demonstrated the safety and efficacy of dietary supplements and natural products. This can reduce the time and cost of developing new drugs [31,32,33].

The primary focus of this review article was to thoroughly analyze and summarize the mechanism by which some natural products and dietary supplements display their therapeutic effect against aflatoxin-induced nephrotoxicity, and to discuss if any of these substances warrant further research.

## 2. Methods

To find research findings on the effect of natural products and dietary supplements on aflatoxin B1-induced nephropathy, articles published in the last decade (from 2010 to 2023) were searched in electronic databases such as PubMed, Google Scholar, and Embase. We identified research articles by performing a preliminary search of abstracts or topics on aflatoxin-induced nephropathy. The following keywords were used in the search: aflatoxin, chronic kidney disease (CKD), renal dysfunction, antioxidant, natural product, dietary supplement. The search was limited to articles published in English. All studies identified in the search were assessed for eligibility and inclusion by two different authors. The inclusion criteria for the review were based on the use of natural products and/or dietary supplements to combat aflatoxin renal toxicity. We excluded studies or original articles that evaluated the effects of natural products and dietary supplements on aflatoxin hepatotoxicity and other mycotoxins, especially ochratoxin.

## 3. Aflatoxin Metabolism and Toxicity

Aflatoxins are classified into six groups: B1, B2, G1, G2, M1 and M2, according to their fluorescence and chromatographic properties [34,35,36]. Aflatoxins are listed as aflatoxin B1 (AFB1)  > aflatoxin G1 > aflatoxin B2  > aflatoxin G2 according to the strength of their toxic effects [37].

AFB1 is the most carcinogenic of the aflatoxins. Cytochrome P450 (CYP) enzymes in the liver, predominantly CYP3A4 and CYP1A2, metabolize AFB1 to the carcinogen aflatoxin B1-8,9-epoxide. In brief, aflatoxin B1-8,9 epoxide binds to the N7 guanine site in DNA and RNA to form AFB1-N7 guanine, which ultimately leads to carcinogenic effects. The harmful metabolite is the primary cause of all toxic effects that inhibit DNA, RNA, and protein synthesis [38,39]. AFB1 is also converted to a number of hydroxylated metabolites, such as aflatoxin M1 (AFM1), the most carcinogenic of the hydroxylated metabolites [39]. Aflatoxin M1 can be detected in dairy products from animals that have consumed food contaminated with AFB1 [7,39]. In addition, contamination of food or poultry feed with fungal toxins reduces poultry safety and quality, causing a huge economic loss at the industrial level [40,41]. The International Agency for Research on Cancer (IARC) has classified AFB1 and AFM1 as group 1 and group 2B human carcinogens, respectively [42]. Acute aflatoxin toxicity is rare in developed countries but common in some developing countries, particularly in Africa, while chronic toxicity is a global problem [6].

### 3.1. Aflatoxin and Renal Dysfunction

The biliary and urinary pathways are the main routes via which AFB1 and its metabolites are excreted. Therefore, AFB1 could be found in the kidney and urine at varying concentrations [43]. It is important to emphasize that research on aflatoxin exposure in populations, and its association with renal effects, is limited. Chronic aflatoxin consumption has been associated with adverse renal effects in animal models. However, evidence in humans is limited. A study to determine the association between AFB1 exposure and early stage renal impairment in indigenous women in San Luis Potosi, Mexico, found a significant association with kidney injury molecule-1 (KIM-1) and cystatin-C (Cys-C) [29]. KIM-1 is expressed in diabetic nephropathy, lithiasis, and sepsis, while Cys-C, a non-glycosylated protease inhibitor protein, is excreted by the kidney and is involved in tubular damage after ischemic kidney injury [44]. AFB1 has been shown in studies to accumulate in the kidney and cause kidney damage by increasing renal function markers including creatinine (CRE), blood urea nitrogen (BUN), and uric acid (UA), all of which are indicators of impaired renal function [45]. Animal models have shown that AFB1 and its metabolites cause oxidative stress, increase inflammation, cell infiltration, and lipid peroxidation, and decrease antioxidant capacity, leading to kidney tissue damage (TD) [2,19,20,21,22,23]. However, the mechanism of AFB1 nephrotoxicity is poorly understood.

### 3.2. Aflatoxin Induces Oxidative Stress in the Kidney

Oxidative stress is an important molecular mechanism for kidney injury in AFB1 nephrotoxicity [46]. It has been reported that AFB1 increases the production of ROS [47]. CYP acts on AFB1 to produce AFB1-8,9-epoxide [48], which is responsible for the induction of oxidative stress of tissues, the depletion of antioxidants, the formation of DNA adducts, and tumor initiation [12]. Oxidative stress appears when the balance between the antioxidant defense system and free radical generation system is disturbed; this might be a cause of several diseases [49]. It can be triggered when the organism is exposed to external harmful stimuli, leading to an increase in various ROS in the body, such as H_2_O_2_, superoxide anion (O_2_^•−^), and hydroxyl radicals (OH·) [9]. ROS can damage proteins, nucleic acids, and other biological macromolecules, and produce large amounts of malondialdehyde (MDA), leading to tissue damage [50]. Therefore, lipid peroxide (MDA) is an important biomarker to assess oxidative damage.

Aflatoxin increases lipid peroxidation and decreases both enzymatic and non-enzymatic antioxidants, leading to oxidative stress [15]. Exposure to AFB1 increases renal MDA and H_2_O_2_ levels [51] and significantly reduces the activity of superoxide dismutase (SOD), reduced glutathione (GSH), catalase (CAT), glutathione peroxidase (GPx), glutathione reductase (GR), and glutathione S-transferase (GST) [19]. AFB1-induced oxidative stress alters the expression of proapoptotic Bax and antiapoptotic Bcl-2, resulting in mitochondria-mediated apoptosis [14,52]. Bax promotes apoptosis by competing with Bcl-2 to enhance cytochrome-c release into the cytoplasm, resulting in the activation of caspase-9/3 cascade, which is hallmark of the mitochondrial apoptotic pathway [53,54]. Caspase-9 can directly activate caspase-3 and induce mitochondrial apoptosis [55]. Therefore, caspase-9 is thought to be an indicator of the mitochondrial apoptotic pathway, whereas cleaved caspase-3 is a key enforcer of apoptosis [51,56,57].

The Kelch-like ECH-associated protein 1 (Keap1) and nuclear factor erythroid 2-related factor 2 (Nrf2) signaling pathway (Keap1/Nrf2) is an effective approach for the body to reduce both endogenous and exogenous oxidative stress through the expression of antioxidant enzymes and phase II detoxification enzymes [49]. In the cytoplasm, Nrf2, an important transcription factor for oxidative stress, is typically coupled to its specific antagonist, Keap1. An oxidative stimulus such as ROS causes Nrf2 to uncouple from Keap1 and translocate to the nucleus, where it binds to the antioxidant response element (ARE) and regulates the expression of downstream genes such as SOD, CAT, GPx, NQO1, and HO-1 [49,58,59]. The Keap1/Nrf2 pathway and its downstream genes are disrupted by AFB1-induced renal oxidative stress, resulting in an increase in oxidative stress metabolites and a decrease in antioxidant enzyme activity [19,60,61,62,63,64]. Therefore, reducing oxidative stress may be a promising therapy against AFB1-induced nephrotoxicity, as shown in Figure 1.

## 4. Natural Products and Dietary Supplements for AFB1-Induced Kidney Damage

Natural products from plant, marine, and botanical sources are widely used globally due to their health benefits and history [65]. Dietary supplements, including vitamins, minerals, proteins, herbs, probiotics, carbohydrates, hormone activators, and oil supplements are intended to improve an individual’s health status and well-being. They are taken during meals or at specific intervals to supplement an individual’s daily diet, especially when there is an imbalance in a nutrient or the absence of a specific nutrient [66]. Therefore, the Food and Drug Administration (FDA) and other international health organizations have established guidelines for the safety and efficacy of dietary supplements for medical use. Although many regulatory bodies restrict dietary supplements and natural products from treating or preventing human disease, a large number of FDA-approved drugs have been developed from natural products [67,68].

The search for nephroprotective drugs with good efficacy and minimal adverse effects is a public health concern. Natural remedies offer multiple pathways, diverse targets, and minimal toxicity, leading to increased research into their use in the prevention and treatment of kidney disease [69]. The literature suggests that natural remedies may reduce AFB1-induced toxicity by increasing antioxidant and anti-inflammatory capabilities and modulating gene expression of detoxifying enzymes, thereby limiting toxicity and tissue damage [16,20,34,47]. Dietary antioxidants derived from medicinal plants play an important role in the treatment of various diseases by boosting the body’s intrinsic antioxidant system. Numerous studies have shown that natural products and dietary supplements effectively control oxidative stress and inflammation by down-regulating pro-inflammatory cytokines and up-regulating antioxidants such as GSH [63]. Natural products and dietary supplements occupy a key position in research against aflatoxins due to their antioxidant activity and ability to act as a precursor of GSH, which binds to aflatoxin-8,9 epoxide and prevents the epoxide from binding to DNA, since the conjugation is an essential detoxification mechanism of aflatoxins [12,62]. This implies that the primary approach to ameliorate the harmful aflatoxin-8,9 epoxide is to increase the levels of intrinsic antioxidants. Furthermore, reduced mitochondrial biogenesis and energetics induced by toxins have been shown to contribute to the aetiology of kidney disease, and natural therapies have been shown to reduce mitochondrial oxidative stress [70]. Natural remedies can enhance renoprotection by activating the Keap1-Nrf2-ARE signaling pathway, as well as sirtuin-1 and sirtuin-3, which promote Nrf2 transcriptional activity and lower mitochondrial ROS levels, respectively [58,59,70].

Over the past decade, Owuni et al. have conducted extensive research on the effects of natural products such as gallic acid [20], caffeic acid [21], apigenin [22], and 3-indole propionic acid [23] against AFB1-induced hepato-renal effects. Other natural products and dietary supplements such as curcumin, resveratrol, diosmin, morin, esculin, fucoidans, selenomethionine, vitamin E, and lycopene, as well as some herbal formulations and probiotics, have also been found to attenuate AFB1-induced kidney damage, as shown in Figure 2 and Table 1.

### 4.1. Curcumin

Curcumin is a phenolic pigment isolated from the rhizome of *Curcuma longa* (turmeric) [19,87], which has been shown in several studies to have antioxidant, antibacterial, inflammatory, and immune-enhancing properties [88,89,90]. It inhibits protein carbonylation, lipid peroxidation, free radical generation, and mitochondrial permeability transition, thereby reducing cell death [91,92,93]. Curcumin powder has three basic components: diferuloylmethane, and its derivatives dimethoxy-curcumin and bisdemethoxy-curcumin [91].

In a study to investigate the possible mechanisms of curcumin protection against AFB1-induced kidney injury, BALB/c mice were challenged with 750 g/kg body weight AFB1 and treated with 200 mg/kg body weight curcumin. AFB1-exposed mice exhibited higher BUN, CRE, and UA levels, as well as renal cell degeneration and necrosis, whereas curcumin treatment improved these biochemical markers. Furthermore, curcumin increased antiapoptotic Bcl-2 expression while inhibiting apoptotic Bax, cytochrome c, and cleaved caspase-3 protein expression in the kidneys of mice, implying that curcumin could reduce AFB1-induced renal apoptosis via the mitochondrial pathway, thereby reducing AFB1-induced nephrotoxicity [19]. Curcumin treatment inhibited keap1 and promoted Nrf2 expression and its downstream genes (SOD1, CAT, NQO1, GCLC, GCLM, and GSH), which were inhibited by AFB1 in the kidney in other studies. In comparison to the AFB1 group, the curcumin intervention group significantly boosted renal antioxidant enzyme activity and also decreased MDA and H_2_O_2_ levels [45]. This finding implies that curcumin might restore AFB1’s disruption of the Keap1-Nrf2 pathway, thereby preventing excessive oxidative damage.

To evaluate the effect of curcumin against AFB1 on the renal cortex of adult male albino rats, AFB1 (250 µg/kg) caused deterioration and necrotic alterations with disruption of the basal lamina. However, curcumin (200 mg/kg b.w.) significantly elevated Bcl-2 expression, decreased glomeruli enlargement with dilation of capillaries, and improved histological and immunohistochemical results in contrast to the untreated group [71]. In another study, curcumin and black tea, which contain polyphenols as potent antioxidants against oxidative hazards [72,94,95], synergistically attenuated nephropathic changes induced by chronic exposure to AFB1 in Sprague Dawley rats. In the study, biochemical and histopathological analysis showed hepatorenal dysfunction in response to AFB1-induction (25 µg/kg b.w). GSH and antiapoptotic Bcl-2 in kidney tissue significantly decreased, while expression of the antitumor p53 increased significantly, as well as lipid peroxidation. Curcumin (200 mg/kg) and black tea (2%), when administered together, significantly ameliorated the alterations caused by AFB1 in the kidney [72]. AFB1 (0.02 mg/kg), used alone or in combination with curcumin (400 mg/kg) to investigate the effects on the renal tissue in poultry, revealed alterations in oxidative stress parameters such as SOD, CAT, GPx, MDA, NOX4 (mRNA and protein expression), and 8-hydroxy-2′-deoxyguanosine (8-OHdG) levels in the kidney of poultry. Curcumin reduced AFB1-induced oxidative stress markers in the chicken kidney, showing promise for the management of aflatoxicosis [39]. Another study by El-Barbary et al. [47] showed that curcumin and garlic can improve renal function markers (CRE, UA, and urea) in Nile tilapia against AFB1.

AFB1-induced oxidative stress contributes to kidney damage, and curcumin’s antioxidant action may help mitigate such adverse effects. Curcumin’s function is its involvement in reducing aflatoxigenic potential by inhibiting CYP isoenzymes, lowering the formation of AFB1-8,9-epoxide, as well as other aflatoxin metabolites [95,96,97].

### 4.2. Resveratrol

Resveratrol is a natural polyphenol present in red wine, rhubarb, and fruits such as blueberries, as well as many red grape varietals, and it is vital for a wide range of biological functions [98]. It has been shown to have many beneficial effects, including anti-inflammatory and antioxidant roles, by increasing the production of antioxidant enzymes [99]. Resveratrol treatment could ameliorate ischemia/reperfusion-induced renal injury and improve renal function by enhancing mitochondrial biogenesis and decreasing ROS [100]. Resveratrol has shown well-proven utility in animal models of diabetic nephropathy, drug-induced injury, ischemia-reperfusion, and sepsis-induced kidney injury [100,101]. Resveratrol ameliorates oxidative stress in the kidney of AFB1-induced hepatocellular carcinoma rats by mediating sirtuin 1 (SIRT1), increasing antioxidant enzymes (SOD, CAT, GPx), and decreasing MDA [73].

Although there are few studies on the toxic effects of AFB2 on human health, it has dangerous effects on various animals [36,102]. Hence, the protective role of resveratrol against AFB2 was investigated in mice. In the study, AFB2 (20 µg/kg b.w.) caused serious changes in selected physiological, biochemical, and cytogenetic parameters, leading to high levels of BUN and CRE as well as MDA and lowered GSH levels in the kidney. Resveratrol treatment (20 mg/kg b.w.) displayed a protective role against these toxic effects [36]. In addition, resveratrol supplementation reduces fatty changes and the accumulation of lymphoid cells with less interstitial spaces in the kidney of birds induced with AFB1 [74]. The protective properties of resveratrol could be attributed to its antioxidant properties [36,100,101].

### 4.3. Gallic Acid

Gallic acid (3,4,5-trihydroxybenzoic acid, GA) is a naturally occurring polyphenolic compound that is an active component of some phytomedicines and herbal drugs. It is found in green tea, red wine, strawberries, pineapples, lemons, gallnuts, sumac, witch hazel, oak bark, and apple peel [103]. GA is known for its potent antioxidant properties and ability to scavenge ROS such as superoxide anion and hydrogen peroxide. GA also has anti-cancer activities [104]. GA’s inhibitory effects on angiotensin-converting enzyme makes it a potential drug candidate for diabetic nephropathy, a major complication of diabetes. Its chelating ability protects cells and tissues from oxidative stress, promoting antioxidant and anti-inflammatory effects while enhancing the regenerative capacity of the liver and kidney [105]. An in vivo finding in rats exposed to AFB1 (75 µg/kg) only, or co-treated with GA (20 or 40 mg/kg), for 28 successive days by Owumi et al. [20] ameliorated AFB1-induced renal dysfunction by reducing oxidative stress and inflammatory markers in the rat kidney. In contrast, co-treatment of rats with GA and AFB1 reduced caspase-3 and caspase-9 activity, suggesting that GA has an anti-apoptotic effect in treated rats. GA was found to reduce AFB1-induced kidney toxicity in rats by significantly increasing endogenous antioxidants, including SOD, CAT, GPx, and GST, as well as total thiol (TSH), thioredoxin (Trx), and thioredoxin reductase (Trx-R). Levels of the anti-inflammatory cytokine IL-10 were also significantly increased. CA significantly reduced renal dysfunction biomarkers (CREA and urea), reactive oxygen nitrogen species (RONS), lipid peroxidation levels, inflammatory markers (NO, myeloperoxidase (MPO) and IL-1β), and xanthine oxidase activities. CA also preserved liver and kidney cytoarchitecture, suggesting its potential as a food additive to mitigate AFB1-induced toxicity. The study did not investigate the role of GA in AFB1 detoxification, but the previous literature suggests that GA inhibits CYP450s involved in AFB1-8-9 epoxide formation and induces phase II drug-metabolising enzymes [104,105]. Therefore, GA may play a role in the detoxification of AFB1 by inhibiting its activation and scavenging AFB1-8-9 epoxide. Further research is required to fully understand the precise mechanisms behind GA-mediated detoxification of AFB1.

### 4.4. Caffeic Acid

Caffeic acid (3,4-dihydroxycinnamic acid) is a polyphenolic secondary metabolite found in various foods, including coffee beans, olives, potatoes, carrots, and fruits. Studies have shown that it has beneficial biochemical properties such as antioxidant, hepatoprotective, nephroprotective, antimicrobial, cardioprotective, and anticancer activity [106,107]. Caffeic acid (CA) acts as an antioxidant by preventing the generation of ROS and reducing oxidative stress by donating electrons to unstable molecules. CA enhances renal mesangial matrix extension, vacuolation, and autophagosome (miRNA) reappearance in diabetic rats fed with a high-fat diet, indicating its potential as a cure for diabetic kidney disease [107,108,109]. CA has also been shown to reduce hepatorenal damage caused by many toxicants, including aflatoxins.

Owumi et al. [21] investigated the efficacy of CA (40 mg/kg) in attenuating AFB1 (50 μg/kg) hepatorenal toxicity in male Wistar rats. CA was found to reduce AFB1-induced kidney toxicity in rats by significantly increasing endogenous antioxidants, including SOD, CAT, GPx, and GST, as well as TSH, Trx, and Trx-R. Levels of the anti-inflammatory IL-10 increased significantly, while IL-1β decreased. CA significantly reduced renal dysfunction biomarkers (CREA and urea), reactive oxygen nitrogen species (RONS), lipid peroxidation and inflammatory markers (NO, MPO, and XO). CA also reduced DNA damage (8-OHdG), apoptosis (caspase-3 and caspase-9), and preserved renal cytoarchitecture, suggesting its potential as a food additive to mitigate AFB1-induced toxicity. This finding suggests that CA can be used as a dietary supplement to mitigate AFB1-induced renal toxicity.

### 4.5. Diosmin

Diosmin (3′,5,7-trihydroxy-4′-methoxyflavone-7-rhamnoglucoside) is a flavonoid glycoside found predominantly in the pericarp of citrus fruits [110,111]. As a flavonoid, it has a wide range of biological activities, including antihyperglycemic, anti-lipid peroxidation, anti-inflammatory, antioxidant, antimutagenic, and antihypertensive properties [111,112,113]. Studies on diosmin have demonstrated its significant role in reducing oxidative stress [114,115,116]. Eraslan et al. [75] assessed the preventive effects of diosmin against oxidative stress caused by aflatoxin in the kidney and the liver as well. In the study, Wistar albino rats were given aflatoxin (500 μg/kg) and treated with 50 mg/kg of diosmin. Diosmin administration to aflatoxin-treated animals resulted in positive changes in antioxidant enzyme activities. The contents of kidney nitric oxide (NO), MDA, and 4-hydroxynonenal (4-HNE), an unsaturated hydroxyalkenal that is produced by lipid peroxidation in cells [116,117] were significantly reduced, while the activities of renal SOD, CAT, and GPx increased compared with the untreated group. There was also a reduction in BUN, CRE, and UA, in the treated group compared to the untreated group [75]. Diosmin is therefore a potential dietary supplement to protect the kidneys from aflatoxin exposure.

### 4.6. Apigenin

Of all the flavonoids, apigenin (4′,5,7-trihydroxyflavone) is one of the most widely distributed compounds in the plant kingdom, and one of the most studied phenolics. It is consumed through diet, as there is currently little evidence of adverse metabolic effects [118]. Apigeninidin-rich fractions of *Sorghum bicolor* extracts have anti-inflammatory and antioxidant properties [118,119,120,121]. *S. bicolor*’s bioactive compounds, 3-deoxyanthocyanidins, which consist of apigeninidin-5-glucoside, apigeninidin, and other derivatives, may help in the management of non-communicable diseases such as diabetes and obesity.

The potential of apigeninidin-rich extracts of *S. bicolor* (5 and 10 mg/kg) to ameliorate AFB1-mediated renal derangements in rats positively reduced CRE and BUN, inhibited oxidative and nitrosative stress, inflammation, and apoptosis, and preserved the histoarchitectural networks of the kidney of AFB1-treated rats. Apigeninidin-rich fractions of *S. bicolor* were found to reduce AFB1-induced toxicity in rats’ kidneys by significantly increasing endogenous antioxidants, including SOD, CAT, GPx, and GST, as well as TSH, Trx, and Trx-R. The levels of anti-inflammatory IL-10 also showed a significant increase, while IL-1β decreased. The fractions further reduced renal dysfunction biomarkers (CREA and urea), reactive oxygen nitrogen species (RONS), lipid peroxidation levels, and inflammatory markers (NO, MPO, and XO activities). It further reduced apoptosis (caspase-3 and caspase-9) and preserved cytoarchitecture in the kidneys of AFB1-treated rats [22]. For the treatment of aflatoxicosis, apigenin and its derivatives could be useful as feed supplements.

### 4.7. Morin

Morin is a natural flavonoid isolated from members of the *Moraceae* family and can be extracted from the stems, fruits, leaves, and branches of many plants [122,123]. Morin has a wide range of pharmacological effects, including protection of DNA from free radical damage, anti-inflammatory properties, and anticancer activity [123,124,125]. It has been shown to significantly reduce CRE and BUN levels in AFB1-induced kidney injury in poultry and to effectively mitigate kidney injury by preventing renal cell necrosis, vacuolization, and exfoliation. Its mode of action has been shown to significantly inhibit MDA while increasing the levels of SOD, GPx, and CAT, and further decreasing inflammatory and apoptosis markers such as tumor necrosis factor-α (TNF-α), interleukin-1β (IL-1β) and interleukin-6 (IL-6), cyclooxygenase-2 (COX-2), inducible nitric oxide synthase (iNOS), and caspase-3, suggesting that morin may be useful in the prevention and treatment of aflatoxicosis [76].

### 4.8. Esculin

Esculin is a glycoside made up of dihydroxycoumarin and glucose. It has been shown to have antimutagenic, anti-inflammatory, and antioxidative effects in recent years [126,127]. Investigations into the protective effect of esculin against pro-oxidant AFB1-induced nephrotoxicity in mice have been conducted by administering AFB1 (66.60 µg/kg) and treating with esculin (150 mg/kg). The protective effectiveness of esculin was determined by evaluating the levels of lipid peroxidation and non-enzymatic antioxidants such as GSH, as well as the activities of enzymatic antioxidants such as GPx, GST, GR, SOD, and CAT in the kidney. Histopathological studies confirmed esculin’s protective effect, as it demonstrated regenerative activity in mouse renal tubules against AFB1-induced nephrotoxicity [77]. The potential of esculin to prevent AFB1-induced nephrotoxicity in mice may be attributed to its antioxidant and free radical scavenging properties.

### 4.9. Fucoidans

Fucoidans are sulfated polysaccharides isolated from the cell walls of various seaweed species, including *Undaria pinnatifida, Sargassum hemiphyllum,* and *Saccharina japonica*, as well as some animal species, including sea cucumber [78,79,128,129,130]. It is high in L-fucose-4-sulfate and low in glucose, uronic acid, galactose, glucuronic acid, rhamnose, xylose, mannose, and arabinose [78]. Fucoidan has recently received a lot of attention from the pharmaceutical sector due to its excellent natural antioxidant properties [64] and effectiveness against inflammation and fibrosis in the kidney [128,129,130,131].

Abdel-Daim et al. [64], evaluated the impact of fucoidan on reducing AFB1-induced hepatorenal toxicity in rats. AFB1 (50 μg/kg, i.p.) increased MDA and NO production while decreasing GSH, GSH-Px, SOD, and CAT activities. There was also an increase in kidney biomarker levels (BUN and CRE) and the overexpression of proliferating cell nuclear antigen (PCNA), which is an indicator of DNA damage [131] in kidney tissues in comparison to the controls. Fucoidan (100 and 200 mg/kg/day, p.o.) consumption alleviated AFB1-induced mitochondrial dysfunction, oxidative damage, and apoptosis. These ameliorated effects are proposed to be attributed to fucoidan’s antioxidant and anti-apoptotic activities [64]. The same author reported a study earlier in which Nile tilapia were given aflatoxin-contaminated feed (2.5 mg/kg diet) and later supplemented with 1% fucoidan. The findings were comparable with those seen in rats [78].

In another study, fucoidan ameliorated BUN and CRE, elevated GSH concentration and GPx, SOD, and CAT activities, and normalized MDA and NO concentrations in renal tissues of aflatoxin-induced diabetic rats [79]. Fucoidan diminishes the accumulation of the extracellular matrix slows down renal fibrosis, glomerulosclerosis, and also expands renal blood flow [131]. It targets multiple signaling systems, including Nrf2/HO-1, NF-κB, ERK and p38 MAPK, TGF-β1, and SIRT1 signaling, which are known to be associated with CKD pathobiology [128,132]. It has been shown to be effective in modifying various risk factors for kidney disease such as, atherosclerosis, dyslipidemia, hyperglycemia, and diabetes [129,130,131]. Fucoidan supplementation could be an efficient protective and therapeutic strategy against AFB1-induced renal injuries and diabetic kidney disease.

### 4.10. Selenium

Selenium (Se) is a trace element that is necessary for human and animal health [133]. It has impacts on DNA repair, the endocrine and immunological systems, and other functions. Due to its great free radical scavenging capacity, selenium can protect the cell membrane and prevent cells from malignant transformation [80,133].

Selenium is highly contained in kidney tissue and can protect the kidney against lipid peroxidation. This is because in CKD, GPx plays an important role in ROS metabolism. Plasma GPx is synthesized in the kidney and requires selenium as a cofactor [134]. Plasma selenium concentration and GPxs enzyme activity in patients with CKD are usually lower than in healthy individuals [135]. GPx production is reduced in CKD and end-stage kidney disease (ESKD), but it is treatable with a selenium supplement. Selenium supplementation has been shown to reduce the severity of malnutrition in haemodialysis (HD) patients by reducing oxidative stress and inflammation [136,137].

AFB1 intoxication results in hepatotoxicity and nephrotoxicity, which led to a reduction in meat and egg production in the poultry industry [41]. AFB1 could induce excessive apoptosis, cell cycle arrest, and cell proliferation depression in poultry [138]. In investigating this, the number of apoptotic renal cells and expressions of Bax and caspase-3 messenger RNA (mRNA) increased significantly, while the expression of Bcl-2 was significantly decreased in a study where poultry birds were administered with AFB1 (0.3 mg/kg). A significantly decreased PCNA expression and arrested G0/G1 phases of the cell cycle were also seen. Selenium (0.4 mg/kg) supplied in the diet, by contrast, improved these parameters, suggesting that aflatoxin could induce apoptosis and cell cycle blockage in the renal cells of poultry [80].

In another study, selenomethionine (SeMet), a naturally occurring amino acid, has been shown to protect Madin-Darby canine kidney (MDCK) cells from 0.25 µg/mL of AFB1-induced oxidative stress by significantly decreasing the MDA level while increasing the intracellular GSH level and GPx activity and significantly improving cell viability [81]. It is therefore presumed that selenium is a potential antioxidative agent to alleviate AFB1-induced oxidative stress, and could be explored to evaluate its ability to inhibit other nephrotoxic agents.

### 4.11. Vitamin E

Vitamin E includes eight fat-soluble compounds (α-, β-, γ-, δ-tocopherol, and α-, β-, γ-, δ-tocotrienol) synthesized in plants and found in fat-rich food and fortified food. The most common and biologically active kind is α-tocopherol [139,140]. Vitamin E is a potent chain-breaking antioxidant that inhibits the production of reactive oxygen species when fat undergoes oxidation and during the propagation of free radical reactions [141]. Alpha-tocopherol safeguards polyunsaturated fatty acids from lipid peroxidation, thereby preserving signaling molecules that are lipid-based and susceptible to oxidation [139]. It acts as the first line of defense against lipid peroxidation, protecting the cell membranes from free radical attack [141,142] and has been shown to be a potent antioxidant, anti-inflammatory, and antifibrotic agent in animal and clinical studies [143].

A study conducted to assess the harmful effect of AFB1 on the renal cortical tissue of rats and the preventive effect of vitamin E concluded that vitamin E alleviates AFB1-induced renal damage. The study showed a significant increase in all the antioxidant markers (GSH, GPx, GR, and GST) and a reduction in MDA as compared to untreated. Alongside the increased oxidative stress production, apoptosis was also induced. The study showed a significant decrease in caspase-3 in the renal cortex homogenate treated with vitamin E as compared to the untreated group. The histological examination of the glomeruli, as well as the renal tubules, appeared nearly intact under both light and electron microscopes [8]. Vitamin E supplementation alleviated nephrotoxicity by correcting oxidative stress parameters and the expression of pro-apoptotic proteins. Hence, Vitamin E could be considered in wide research studies to be a first-choice antioxidant supplement due to its cost-effectiveness and ability to prevent fungal toxins and pro-oxidant-induced renal dysfunction.

### 4.12. 3-Indole Propionic Acid

Dysbiosis influences the bacterial-mediated production of indole-containing uremic toxins, such as indoxyl sulfate and p-cresol sulfate, which are derived from tryptophan [144]. Endogenous 3-indole propionic acid (3-IPA) is a metabolite of tryptophan metabolized by the gut microbiota that can prevent redox imbalance, inflammation, and cellular lipid damage [145]. Clinical studies show that 3-IPA levels decrease significantly in patients with renal impairment, while indoxyl sulfate and p-cresol sulfate levels increase. This suggests that 3-IPA may be an important biomarker and renal protector against the development of kidney disease [144,145]. The beneficial effect of 3-IPA (25 and 50 mg/kg) against AFB1-mediated (50 µg/kg) organ toxicity in rats has been assessed in a study where 3-IPA supplementation abated biomarkers of renal dysfunction in rat serum. Antioxidant mediators and enzymes (GSH, SOD, CAT, GPx, and GST) increased while pro-oxidative biomarkers, including lipid peroxidation, reactive oxygen nitrogen species (RONS) and 8-OHdG, reduced significantly in the 3-IPA treatment group. Kidney pro-inflammatory and apoptotic mediators, including xanthine oxidase (XO), myeloperoxidase (MPO), NO, IL-1β, and Caspases 9 and 3, were suppressed and interleukin 10 (IL-10) upregulated. 3-IPA further ameliorated CRE and BUN and attenuated glomerular atrophy [23]. These data suggest that increasing endogenous 3-IPA may be a potential method to prevent xenobiotic-mediated kidney damage, particularly AFB1.

### 4.13. Lycopene

Lycopene is a naturally occurring carotenoid that can be found in tomatoes, watermelon and other foods [146,147,148]. It has a wide range of biological activities, including protection against cancer, oxidation, aging, and inflammation [149,150,151]. Lycopene is an effective antioxidant and has a high potential for scavenging free radicals [152]. A cohort study found that higher serum lycopene levels reduced CVD mortality and CKD due to lycopene’s ability to scavenge ROS [153].

In a study to assess the protective effect of lycopene against the deleterious effects of AFB1 exposure in the kidneys of Wistar Albino rats by measuring antioxidant defense systems, rats were administered low and high doses of aflatoxin (0.5 mg/kg/day and 1.5 mg/kg/day, respectively). MDA levels increased significantly, while GSH, GST, CAT, GPx, SOD, and G6PD activities decreased in the kidneys of the untreated group. There was also an increase in BUN and CRE levels, and a reduction in sodium concentrations in the plasma of AFB1-treated rats. Treatment with lycopene (5 mg/kg/day) compensated for the AFB1-induced decrease in renal GSH, CAT, GPx, GST, SOD, and G6PD. In addition, lycopene showed protection against aflatoxin-induced nephrotoxicity by ameliorating BUN and CRE levels, which may be due to its antioxidant activity [82]. Lycopene has attracted much attention for its antioxidant effect in activating Nrf2, a key transcription factor responsible for regulating the cellular oxidative stress response [154,155,156,157], which may alleviate many types of nephropathies, such as type 2 diabetes mellitus, CKD, and acute kidney injury (AKI) [152,153].

El-Sheshtawy et al. [83] also conducted a study in which ducklings were fed 30 ppb aflatoxin for 2 weeks, followed by treatment with 100 mg/kg lycopene or 600 mg/kg silymarin for 10 days. Silymarin is a phytochemical (flavonoid) with antioxidant properties that can regenerate liver cells, suppress lipid peroxidation, and provide anti-inflammatory benefits [158]. In the study, kidney parameters were higher in the aflatoxin-poisoned group and lower in the lycopene and silymarin treated groups. The untreated group showed high MDA and aflatoxin residues with worse antioxidant indices (TAC, GST and CAT). At the same time, treatment with lycopene or silymarin reversed the effect of aflatoxin on MDA and increased antioxidant activity [83]. Lycopene [159] and silymarin [158], which have high antioxidant activity, can be used to counteract the damaging effects of aflatoxin on kidney tissue.

## 5. Herbal Product

Medicinal plants have a wide spectrum of natural antioxidants and are used to cure a wide range of ailments worldwide [160] due to the synergism that occurs among phytochemicals in the plants [161]. Chinese herbal medicines are widely used in China and some Asian countries to treat chronic diseases such as CKD, but there is a lack of high-quality clinical evidence worldwide. In 2015, Lin et al. [31] conducted the first population-based study on the use of Chinese herbal medicines in patients with CKD. The study found that patients who received herbal medicines had a significant (60%) reduction in the risk of ESRD (*n* = 24,971 patients). The data provide strong evidence that the use of herbal medicines could reduce the incidence of ESRD in CKD patients. The authors did not provide a comprehensive explanation of the exact mechanism behind the reduction. However, Chinese herbal medicine has been proven to have anti-inflammatory, antioxidant, and immunomodulatory properties against kidney disease [32,162]. Therefore, Kalpaamruthaa herbal formulations and fenugreek, which have similar properties, have been tested in animals for their ability to attenuate aflatoxin-induced nephrotoxicity.

### 5.1. Kalpaamruthaa

Kanchana et al. [30] investigated the therapeutic effect of Kalpaamruthaa (KA), an indigenous modified Siddha formulation consisting of *Semecarpus anacardium* nut milk extract, *Emblica officinalis* fruit and honey. The nut milk extract of *S. anacardium* Linn., which is the main component of KA, has numerous therapeutic applications due to its phenolic compounds. It has been shown in clinical studies to have antioxidant, anti-inflammatory, hypoglycemic, antiarthritic, immunomodulatory, anticancer, and antidiabetic activities [163,164]. The *Emblica* fruit has been shown to contain constituents with variable biological activity, such as vitamin C and tannins, and is a good source of phenolics. It has many pharmacological activities, such as antioxidant, anti-diabetic, cytoprotective, hepatoprotective, and anti-tumor activities [165]. Honey is rich in antioxidant, antiviral, antifungal, antibacterial, antiseptic, and anticarcinogenic properties [166].

The research group investigated the therapeutic effect of KA (200 mg/kg/day) against AFB1 (2 mg/kg) induced in rat kidneys. Oral treatment of KA showed significant ameliorative effects on all the biochemical studies, including BUN, CRE, renal marker enzymes, MDA, and increased levels of tissue antioxidants SOD, CAT, GPx, GR, and GSH in response to AFB1 administration [30]. The findings indicate that KA treatment alleviates AFB1-induced oxidative stress in renal tissues due to the antioxidant properties of the composition. The components of KA may have worked synergistically to produce such an effect. KA can therefore be standardized, validated, and studied for the treatment of kidney disease.

### 5.2. Fenugreek

Fenugreek (*Trigonella foenum* graecum L.) is a member of the Fabaceae family that has been used in folk medicine for centuries to treat a variety of ailments [167,168]. It exhibits antioxidant and antimicrobial functions, anti-cancer activity, anti-diabetic, anti-atherosclerotic, immunomodulatory, and anti-aflatoxin effects [169,170,171]. Fenugreek has been shown to exert hepatoprotective and nephroprotective effects by reducing lipid peroxidation levels in the liver and kidney, in addition to enhancing cellular antioxidant defense in rats [172,173,174]. To evaluate the effect of fenugreek on AFB1-induced damage to the liver and kidney, male albino mice received AFB1 (20 µg/kg) followed by fenugreek (200 mg/kg/day) [84]. AFB1 caused liver and kidney damage in addition to alterations in some hematological parameters that were manifested by increased oxidative damage, a substantial reduction in antioxidant capacity, and increased kidney histopathological findings. A significant rise in BUN, UA, and CRE levels was observed in the aflatoxin group compared to the control mice. In the fenugreek groups, a significant decline was observed in BUN, UA, and CRE levels, together with an improvement in kidney histological and cellular structures. In addition, the treatment ameliorated the levels of MDA, total antioxidant enzyme activities, and GSH in the kidney and liver [84]. In this regard, fenugreek seed supplementation could be recommended as a regular nutrient for the liver [175] and for kidney protection against aflatoxin toxicity and other nephrotoxic agents.

## 6. Probiotics

In recent years, a great number of studies have focused on the impacts of intestinal microbiota on an individual’s health status [138,176,177]. Probiotics are known to have many beneficial health effects, and the consumption of probiotics alone or in food shows that strain-specific probiotics can enhance antioxidant activity and reduce damage caused by oxidation [178]. Probiotics alleviate cytotoxicity and inflammation induced by AFB1 exposure [85,86,179]. *Bacillus amyloliquefaciens,* one of the most widely used probiotics, has been shown to alleviate AFB1-induced kidney injury in mice [9,55]. AFB1 (450 µg/kg) administration showed biochemical indices and pathological alterations, as well as an increase in apoptotic cells in the kidney tissue of mice. After *B. amyloliquefaciens* B10 (3.5 × 10^9^ CFU/mL) treatment, serum UA, BUN, CRE, and TUNEL-positive cells significantly decreased; GPx, SOD, and CAT increased, while MDA reduced significantly. Protein expressions of Nrf2, HO-1, AKT, p-AKT, and Bcl-2 significantly increased, while Keap-1, PTEN, Bax, Caspase-9, and Caspase-3 significantly decreased. This evidence confirms that the use of *B. amyloliquefaciens* B10 (as a probiotic) alleviates AFB1-induced kidney injury in mice [9]. Probiotic preparations containing *Lactobacilli reuteri, plantarum, pentosus, rhamnosus, paracasei*, and *Saccharomyces cerevisiae* have also been shown to ameliorate AFB1 residues and alleviate histopathological changes in poultry kidneys [85]. It can be inferred that probiotic supplementation can help mitigate the adverse effects of AFB1 contamination in poultry feed, thereby improving bird health and product safety.

## 7. Conclusions and Perspectives

Chronic aflatoxin consumption has been associated with renal toxicity in animals [180], but data in humans are limited. Recently, a study found a significant association between aflatoxin exposure and early stage renal impairment in humans [29]. This warrants the search for novel therapeutic agents for aflatoxin-induced nephrotoxicity and kidney-related diseases such as AKI and CKD. Many patients with kidney disease rely on natural therapies; however, the mechanisms of action of these therapies are poorly understood due to the numerous pathways and targets they provide. Clinical studies have shown that natural products and dietary supplements have antioxidant properties that could manage CKD in patients. Animal models have confirmed their efficacy in reducing renal apoptosis, DNA damage, intertubular hemorrhage, dilatation, and cellular antioxidants. This study has so far highlighted some active compounds, crude extracts, herbal formulations, and probiotics with various bioactive ingredients against aflatoxin-induced renal dysfunction and illuminated their mechanisms of action in both in vitro and in vivo studies, which could be further explored. The natural products and dietary supplements mentioned in this review target renal oxidative stress, inflammation, tissue damage, and kidney function markers induced by aflatoxin, and could be considered in wide experimental studies for the prevention of renal damage induced by fungal toxins and other nephrotoxic agents. Our findings, together with those of Bbosa et al. [12], suggest that the protective effect of these natural products and dietary supplements against AFB1 toxicity may be due to their ability to act as precursors of GSH, which binds to aflatoxin-8,9 epoxide, and prevents the epoxide from binding to DNA since the conjugation is an essential detoxification mechanism of aflatoxin. The information provided in this study may improve the management of kidney disease associated with mycotoxin exposure, and potentially aid in animal feed supplementation. However, in order to translate the results of this study into therapeutic applications for patients with kidney disease, a more detailed characterization of the molecular targets of natural products and dietary supplements is required.

## Figures and Tables

**Figure 1 ijms-25-02849-f001:**
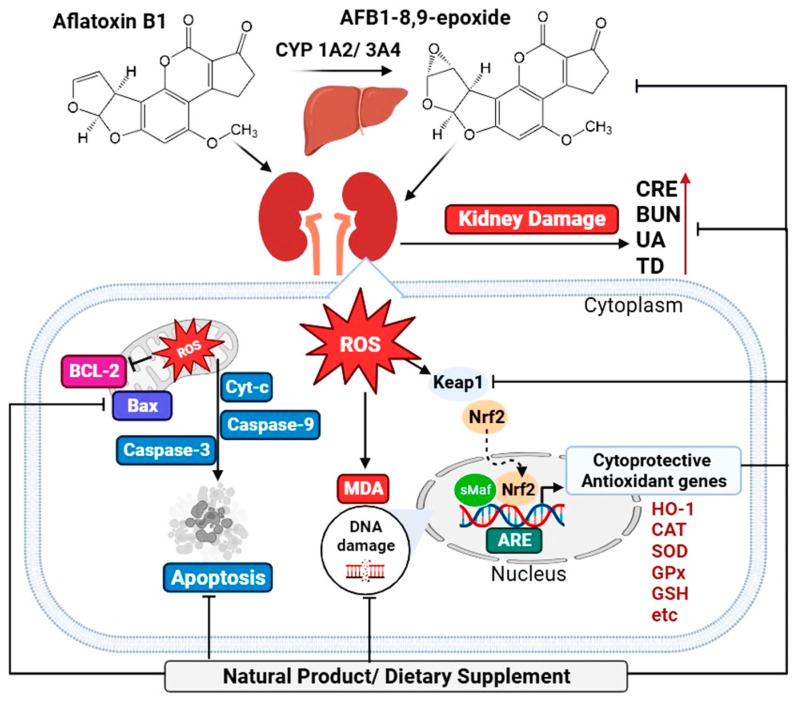
Schematic representation of the mechanism of action of natural products and dietary supplements against AFB1-induced nephrotoxicity. AFB1 and its metabolite, AFB1-8,9 epoxide, which are metabolized in the liver, increase ROS and induce kidney tissue damage, thereby increasing renal function markers. The ROS-generated induce mitochondrial apoptosis by inhibiting the anti-apoptotic BCl-2 and promoting the pro-apoptotic Bax, cytochrome-C, and caspase 9 and 3. AFB1 also interferes with the Keap1/Nrf2 pathway and its downstream genes. The natural products and dietary supplements mentioned in this review may reverse the effects of AFB1 on kidney function. The image was created using BioRender.

**Figure 2 ijms-25-02849-f002:**
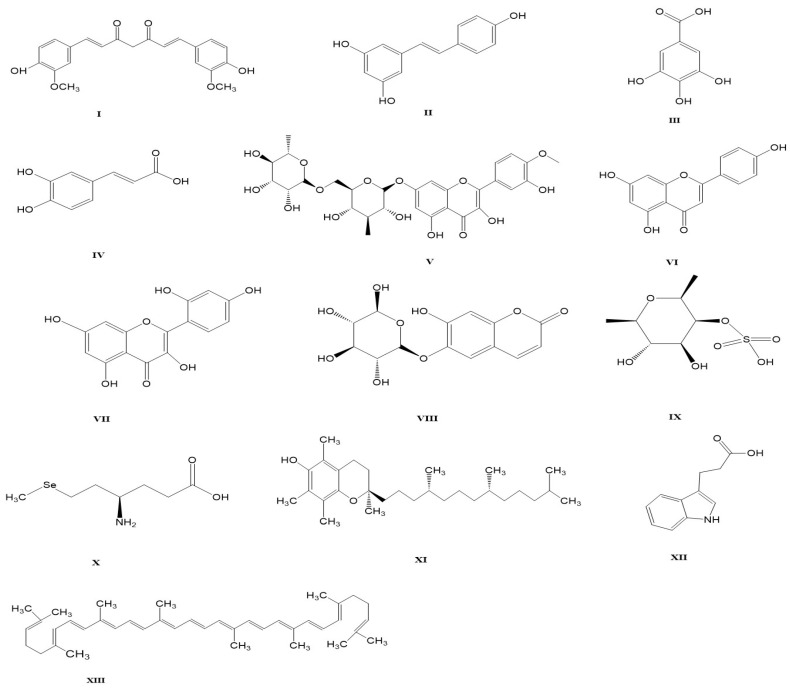
Chemical structures of natural products and dietary supplements against AFB1-induced nephrotoxicity. (**I**) Curcumin, (**II**) Resveratrol, (**III**) Gallic acid, (**IV**) Caffeic acid, (**V**) Diosmin, (**VI**) Apigenin, (**VII**) Morin, (**VIII**) Esculin, (**IX**) Fucoidans, (**X**) Selenomethionine, (**XI**) Vitamin E, (**XII**) 3-indole propionic acid, (**XIII**) Lycopene. The compounds listed are classified as phenolics (**I**–**IV**), flavonoids (**V**–**VII**), coumarin glucoside (**VIII**), polysaccharide (**IX**), amino acid (**X**), vitamin (**XI**), tryptophan metabolite (**XII**), and carotenoid (**XIII**). The chemical structures were sketched using ChemDraw Professional Software Ultra 22.2 PerkinElmer (CambridgeSoft, Cambridge, MA, USA).

**Table 1 ijms-25-02849-t001:** Effect of natural products and dietary supplements on aflatoxin-induced renal dysfunction.

Natural Products/Dietary Supplements	Model	Effect/Mechanism	Ref.
Curcumin		Reduces renal function markers:	
	Mice	↓ Urea, CREA, UA	
		Gene and protein expression associated with the Keap1/Nrf2 pathway:	
		↑ SOD1, CAT, NQO1, GSH, GCLC, GCLM, Nrf2	[19]
		↓ Keap1	
		Kidney antioxidant capacity:	
		↑ SOD, CAT, H_2_O_2_, GSH, T-AOC	
		↓ MDA	
		Renal Apoptosis: ↓ Bax, Cyt-c, Caspase-3, Caspase-9, TUNEL-positive cells	
		↑ BCl-2	
		Restored pathological structure of kidney.	
	Rat	Reduction in glomeruli enlargement and histological improvement.	[71]
		↑ Bcl-2	
	Poultry	↑ SOD, CAT, GPx	[39]
		↓ MDA, 8-OHdG, NOX4	
Curcumin and black tea	Rat	↓ MDA	[72]
		↑ GSH and Bcl-2 in kidney tissue	
		Anti-tumor p53	
Resveratrol		Reduces renal function markers and excessive oxidative damage.	
	Mice	↑ GSH	[36]
		↓ MDA	
		↓ Urea, CREA	
	Rat	↑ SOD, CAT, GPx	[73]
		↓ SIRT1,	
		↓ MDA	
		↓ UA	
	Poultry	↓ Fatty kidney	[74]
		↓ Interstitial spaces of kidney	
Gallic Acid	Rat	Improves renal in kidney cytoarchitecture, such as glomerular mesangialization and inflammatory cell infiltration.	[20]
		↑ CAT, SOD, GPx, GST, GSH	
		↓ RONs, LPO, NO, MPO,	
		↓ Renal caspase 3, TNF-α,	
		↓ Urea, CREA	
Caffeic acid	Rat	Improvements kidney cytoarchitecture, such as glomerular mesangialization and inflammatory cell infiltration.	[21]
		↑ Kidney weight	
		↓ Urea, CREA	
		↑ SOD, CAT, GPx, GST	
		↑ GSH, TSH and Trx l	
		↓ XO, RONS, LPO	
		↓ NO, MPO, IL-1β, IL-10	
		↓ 8-OHdG, caspase-3, caspase-9	
Diosmin	Rat	Reduces renal function markers, and renal oxidative stress.	[75]
		↓ MDA, 4-HNE, NO	
		↑ SOD, CAT, GPx	
		↓ BUN, CREA, UA	
Apigenin	Rat	Anti-inflammatory, reduces renal function markers, and preserves the histoarchitectural network of kidney.	[22]
		↓ Urea, CREA	
		↑ SOD, CAT, GPx, GST	
		↑ GSH, TSH and Trx l	
		↓ XO, RONS, LPO	
		↓ NO, MPO, IL-1β, IL-10	
		↓ caspase-3, caspase-9	
Morin	Poultry	Effectively relieved kidney damage; Renal cell necrosis, exfoliation, and vacuolization.	[76]
		↓ BUN, CREA	
		↑ SOD, GPx, CAT	
		↓ MDA	
		↓ TNF-α, IL-6 IL-1β iNOS COX-2	
		↓ caspase 1, 3 and 11	
Esculin	Mice	Exhibits regenerative activity in renal tubules and antioxidant activity.	[77]
		↑ GST, GSH, GPx, GR, CAT, SOD	
		↓ MDA	
Fucoidan	Rat/Nile Tilapia	Improves renal function markers and tissue damage.	[64,78]
		↑ Nrf2, HO-1, CAT, SOD, GPx, GST	
		↓ MDA, NO, DNA Damage (PCNA)	
		↓ BUN, CREA	
	Diabetic Rat	↑ SOD, CAT, GSH, GPx	[79]
		↓ MDA	
		↓ BUN, CREA	
		↓ 8-OHdG	
		↓ IL-1β, IL-6, TNF-α	
Selenium	Poultry	Improves renal apoptosis and DNA damage.	[80]
		↑ Bcl-2	
		↓ Cell cycle blockage in renal cell	
		↓ DNA damage (PCNA), Caspase-3	
	MDCK cells	↓ Cytotoxicity	[81]
		↓ MDA	
		↑ mRNA GPx, GSH	
Vitamin E	Rat	Reduces glomerular architectural impairment and blood renal barriers in the renal cortex.	[8]
		↓ Renal caspase 3	
		↑ GSH, GPx, GR, GST	
		↓ MDA	
		↓ Urea, CREA	
3-indole propionic acid	Rat	Moderate glomerular atrophy	[23]
		↑ GST, GSH, SOD, CAT, GPx, TSH	
		↓ XO, RONS, LPO	
		↓ MDA, 8-OHdG	
		↓ Caspase-9, and 3	
		↓ CREA, BUN	
		↓ NO, IL-1β	
		↑ IL-10	
Lycopene	Rat	Improves intertubular hemorrhage and dilatation in tubules.	[82]
		↑ GSH, GST, GPx, CAT, SOD, G6PD	
		↓ BUN, CREA, UA	
		↓ MDA	
Lycopene or silymarin	Ducklings	↑ SOD, CAT, TAC, GST	[83]
		↓ MDA	
		↓ Urea, CREA	
Kalpaamruthaa (*S. anacardium* and *E. officinalis*)	Rat	↑ CAT, SOD, GR, GPx, GSH	[30]
		↓ MDA	
		↓ BUN, CREA	
Fenugreek	Mice	Improvement of kidney histological and cellular structure.	[84]
		↑ GSH	
		↓ CREA, Urea, UA	
		↓ MDA	
Probiotic	Mice	Activates antioxidant response element and inhibits renal apoptosis.	[9]
		↑ Nrf2, HO-1, AKT, P-AKT, BCl-2	
		↑ SOD, CAT, GPx	
		↓ Keap1, PTEN, Bax, Caspase 9 & 3	
		↓ UA, CREA, TUNEL positive cells	
		↓ MDA	
	Poultry	Reduces aflatoxin levels in kidney.	[85,86]

↑ Increase/Promote; ↓ Decrease/Reduce. Natural products and dietary supplements mentioned in this review mainly regulate renal oxidative stress, inflammation, tissue damage, and kidney function markers induced by aflatoxin.

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
