# Peer review of "Therapeutic Effect of Natural Products and Dietary Supplements on Aflatoxin-Induced Nephropathy"

_ijms, 2024, doi:10.3390/ijms25052849_

Round 1
Reviewer 1 Report
Comments and Suggestions for Authors
The review article is original and very interesting.
The topic is very relevant, the aim of this review being to highlight the active compounds, crude extracts, herbal formulations, and probiotics against aflatoxin- induced renal dysfunction by illuminating their mechanisms of action.
Methodology is very good, authors summarizing the research findings on the effect of natural products and dietary supplements on aflatoxin B1-induced nephropathy.
The results have revealed that natural products and dietary supplements mentioned targeted renal oxidative stress, inflammation, tissue damage, and kidney function markers induced by aflatoxin, which could be considered in wide experimental studies in the prevention of fungal toxins-induced renal impairment. The protective effect of these natural products and dietary supplements against AFB1 toxicity could be attributed to their ability to act as a precursor of GSH, which combines with aflatoxin-8,9 epoxide and prevents the epoxide from binding to DNA, since the conjugation is an essential detoxification mechanism of aflatoxins.
The conclusions are consistent with the evidence and arguments presented.
The references are very relevant, including also some relevant author’s previous experience in the field.
I suggest some small corrections
1. The Abstract could be more detailed, to evidentiate the main findings of the article
2. References could be improved, to include some other natural products involved in detoxification of aflatoxins.
See also
1.Carmen Solcan, Mihaela Gogu, Viorel Floristean, Bogdan Oprisan, and Gheorghe Solcan, The hepatoprotective effect of sea buckthorn (Hippophae rhamnoides) berries on induced aflatoxin B1 poisoning in chickens, Poultry Science, 2013; 92(4):966-974, doi: 10.3382/ps.2012-02572, http://ps.fass.org/content/92/4/966.full
2.Bondar, A.; Horodincu, L.; Solcan, G.; Solcan, C. Use of Spirulina platensis and Curcuma longa as Nutraceuticals in Poultry. Agriculture 2023, 13, 8, 1553. https://doi.org/10.3390/agriculture13081553
Reviewer 2 Report
Comments and Suggestions for Authors
In this review the authors show a large number of imformations about the use of natural products against the renal toxicity induced by Aflatoxin B1. The review is very interesting and it could be a valid help for the scientific community.
However some suggestions are requested:
Line 35: renal functions in case of renal failure are compromised regardless of Aflatoxin. The sentence is ambiguous. Remove or modify it
Line 45: modern medicine has found and continues to find important treatments for the treatment of kidney failure, obviously the cure depends on the cause. Written in the proposed way it seems that in all cases there is no suitable treatment, not only for aflatoxin. Please change the concept
Line 79: what evidence suggests that it also accumulates in the kidney? It is not clear the metabolism and distribution. Learn more about it.
Line 231: information on galic acid could be increased
Line 258: the considerations of the effect of epigenin are too general
Line 411: herbal product very superficial information. Go into detail
Reviewer 3 Report
Comments and Suggestions for Authors
The manuscript entitled: ”Therapeutic Effect of Natural Products and Dietary Supplements on Aflatoxin-induced Nephropathy” reports in form of a review data on the Aflatoxins hepatotoxic effect. The topic is well known and exploited in the literature. There are a few drawbacks regarding the proposed manuscript, in particular the novelty and impact on the area of interest. The methodological approach should be more detailed. The relationship between the dietary supplement and nephropaty should be stressed as per the manuscript title. The mentioned substances (maninly natural substances) can be part of a dietary supplement (e.g. lycopene, etc.): please detail better this point. More data on therapeutic effect should be summarized to assess the scope and novelty of the manuscript. In the present form it should not be considered for publication in the Journal.
Comments on the Quality of English LanguageThe quality of English language seems fine: ony minor/moderate editing would be required.
Round 2
Reviewer 1 Report
Comments and Suggestions for Authors
The article was improved significantly according to reviewers suggestions. I recommend to be accepted in present form
Author Response
Thank you very much for taking the time to review this manuscript.
All spelling and grammatical errors have been corrected throughout the manuscript. We also rearranged some of the sections in the track change/ highlighted in yellow.
In the current revision, we have elaborated on the methods [Page 3; line 95].
We also rearranged some of the sections in the track change/ highlighted in yellow.
We have revised the entire conclusion, setting out our findings and the way forward.
Reviewer 3 Report
Comments and Suggestions for Authors
The manuscript entitled: “Therapeutic Effect of Natural Products and Dietary Supplements on Aflatoxin-induced Nephropathy”, yet interesting, still presents some points to address better. The methodology used in the Review should in first be described better and be not subjective (selection criteria for including or excluding studies from the proposed Review paper). The reported data refer mainly to studies on animal please this should be somehow included in the title of the manuscript. Section number four should better substantiate the role of supplements in mitigating the toxic effect of mycotoxins in kidney damage. The mechanism of action should be detailed and commented better. Table 1 reports results with reference to natural substances: please detail which supplements have been evaluated in the Review proposed. One single compounds is considered a supplement? Is there any study on humans assessing the beneficial function of suplements against kidney damage? Please comment on this point. The mechanism of action should be exploited and deetails given (see lines 568 and following). At line 577 the amount range could be an useful information to add to assess the context.
Comments on the Quality of English LanguageThe English language seem fine, minor editing required.
